# SARs for the Antiparasitic Plant Metabolite Pulchrol. 3. Combinations of New Substituents in A/B-Rings and A/C-Rings

**DOI:** 10.3390/molecules26133944

**Published:** 2021-06-28

**Authors:** Paola Terrazas, Efrain Salamanca, Marcelo Dávila, Sophie Manner, Alberto Gimenez, Olov Sterner

**Affiliations:** 1Department of Chemistry, Centre for Analysis and Synthesis, Lund University, 22100 Lund, Sweden; paola.terrazas_villarroel@chem.lu.se (P.T.); Sophie.Manner@chem.lu.se (S.M.); 2Centre of Agroindustrial Technology, San Simón University, Cochabamba 3299, Bolivia; marcelodavila@fcyt.umss.edu.bo; 3Institute for Pharmacological and Biochemical Sciences, San Andrés University, La Paz 3299, Bolivia; efrain_salamanca@hotmail.com (E.S.); agimenez@megalink.com (A.G.)

**Keywords:** *Trypanosoma cruzi*, *Leishmania braziliensis*, *Leishmania amazonensis*, pulchrol, pulchral, cannabinol, SARs

## Abstract

The natural products pulchrol and pulchral, isolated from the roots of the Mexican plant *Bourreria pulchra*, have previously been shown to possess antiparasitic activity towards *Trypanosoma* *cruzi*, *Leishmania braziliensis* and *L. amazonensis*, which are protozoa responsible for Chagas disease and leishmaniasis. These infections have been classified as neglected diseases, and still require the development of safer and more efficient alternatives to their current treatments. Recent SARs studies, based on the pulchrol scaffold, showed which effects exchanges of its substituents have on the antileishmanial and antitrypanosomal activity. Many of the analogues prepared were shown to be more potent than pulchrol and the current drugs used to treat leishmaniasis and Chagas disease (miltefosine and benznidazole, respectively), in vitro. Moreover, indications of some of the possible interactions that may take place in the binding sites were also identified. In this study, 12 analogues with modifications at two or three different positions in two of the three rings were prepared by synthetic and semi-synthetic procedures. The molecules were assayed in vitro towards *T. cruzi* epimastigotes, *L. braziliensis* promastigotes, and *L. amazonensis* promastigotes. Some compounds had higher antiparasitic activity than the parental compound pulchrol, and in some cases even benznidazole and miltefosine. The best combinations in this subset are with carbonyl functionalities in the A-ring and isopropyl groups in the C-ring, as well as with alkyl substituents in both the A- and C-rings combined with a hydroxyl group in position 1 (C-ring). The latter corresponds to cannabinol, which indeed was shown to be potent towards all the parasites.

## 1. Introduction

Natural products have been one of the main sources for bioactive compounds used to treat a wide range of diseases [1]. The chemical diversity, potential selectivity, and the availability of traditional knowledge about the use of natural materials, played an important role in the development of modern drugs. Some examples are morphine isolated from opium [2]; taxol isolated from *Taxus brevifolia* [3]; important antiparasitic drugs, such as quinine, isolated from *Cinchona officinalis* [4]; and artemisinin extracted from *Artemisia annua* [5].

Natural products are often isolated in limited quantities and can be assayed in just a few biological systems. In order to extend the bioactivity scope, synthetic routes can be designed to obtain sufficient quantities of the product, and can also be used to prepare new derivatives and analogues, which may possess superior biological properties [1,6]. For diseases in which the mechanism of action and the drug target are not fully understood, as is the case for leishmaniasis and Chagas disease [7,8,9], derivatives can be prepared and assayed towards cells or whole organisms to measure their activity [10,11,12,13], leading to the development of structure–activity relationships studies (SARs). These can be used to design more potent and less toxic analogues, assuming that just one target is involved [11,14].

The vegetal specie, *Bourreria pulchra*, which is native in the Yucatan province in Mexico, is traditionally used to treat cutaneous diseases, fevers, and infections [15]. The isolation of the main chemical compounds from its roots yielded pulchrol (**1a**, see Figure 1), which was shown to possess antiparasitic activity against *Leishmania braziliensis*, *L. amazonensis* and *L. mexicana* promastigotes, as well as against *Trypanosoma cruzi* epimastigotes [16]. 

*Leishmania* and *Trypanosoma* parasites are part of the family Trypanosomatidae, and are responsible for leishmaniasis and Chagas disease, respectively. Both are considered neglected diseases, but still affect millions of people in developing countries, mainly placed in the tropical and subtropical regions of the world [7,17,18,19,20,21]. Leishmaniasis can appear as cutaneous, mucocutaneous or visceral leishmaniasis, and around 700,000 to 1 million new cases are diagnosed every year. The treatments for leishmaniasis (mainly amphotericin, miltefosine and pentavalent antimonials) may give several toxic side effects and may require hospitalization [22,23]. Likewise, Chagas disease is able to produce damage in the hearth tissue and eventually cause death. Twelve million people are affected by this disease and the existing treatments are far from ideal (nifurtimox or benznidazole) [24,25,26,27,28]. Currently, there are few validated drug targets for leishmaniasis and Chagas disease, and too little is understood of the complex life cycle of these pathogens [9].

Pulchrol (**1a**) is one example of the many natural products that are based on a 6*H*-benzo[*c*]chromene scaffold. This type of compound has been shown to possess different kinds of biological activities [29,30,31,32,33,34]. The most studied benzo[*c*]chromenes are probably the cannabinoids, isolated mainly from the plant *Cannabis sativa*, and known for their affinity to the cannabinoid receptors CB1 and CB2. The natural product cannabinol (**5f**, see Figure 1), which possess the same skeleton as pulchrol, has been shown to be selective for CB2, a receptor expressed on immune cells, macrophages, and in other peripheral organs, while Δ^9^-tetrahydrocannabinol (THC, **6**, see Figure 1), also isolated from *C. sativa*, showed greater affinity for CB1, which is the receptor associated with the psychotropic activity of *C. sativa* [35]. SARs including cannabinol (**5f**) have been studied previously, and indicated that hydroxyl groups at positions 1 and 1′, together with bulky alkyl substituents at position 3, are important for the affinity to CB1 [36]. In addition, shorter and bulkier alkyl substituents at C-3 improved the affinity for the CB2 receptor [37,38]. Cannabinoids have also shown immunomodulatory properties in the treatment of psoriasis [39], antinociceptive properties related to their capacity to induce vasorelaxation and release neuropeptides [40,41], and antineoplastic activity on Lewis lung tumors [33].

The potential shown by pulchrol (**1a**) as an antiparasitic agent against *T. cruzi* and the *Leishmania* species led to the development of a synthetic route [42,43] that yields sufficient amounts of pulchrol to perform additional biological assays. This could also be adapted for the preparation of a series of analogues with individual transformations of the benzyl alcohol moiety in the A-ring [44], variations at positions 1, 2 and 3 in the C-ring, and modifications at the only available position in the B-ring (C-6) [45]. In this investigation, we prepared analogues with combinations of different functionalities in the A-, B-, and C-rings (see Figure 2), inspired by some of the SARs observed in the previous studies [44,45]. Cannabinol (**5f**) and its 3-methyl analogue **5e** (see Figure 2) were also prepared and assayed. Our main objective was to study the effect that several substituents coexisting at different positions in the benzo[*c*]chromene skeleton may have in the activity towards the parasites under study, and retrieve more information about the chemical surroundings of the active site in which pulchrol and its analogues may be interacting.

## 2. Results

### 2.1. Preparation

In this study, analogues of pulchrol were prepared, containing combinations of two or three modifications in the A-, B, and C-rings (see Figure 2). The synthetic routes used to prepare the analogues were partially based on the already reported procedure that was used to prepare pulchrol from biaryl intermediates [42,43]. The molecules in the **1**-series (see Figure 2) were previously reported [44,45], and were used as the starting material to prepare the compounds from the **2**-, **3**-, **4**- and **5**-series, containing a 1′-aldehyde, 1′-methyl ketone, 3-methylbutanoic acid ester, and 9-methyl functionalities, respectively (see Figure 2). Compound **5f** (cannabinol) and its analogue **5e** were obtained through iodine-mediated deconstructive annulation in a one-pot synthesis, using citral and resorcinol analogues as the starting material [46]. In total, 12 molecules were prepared and assayed in vitro against *T. cruzi* epimastigotes, as well as *L. braziliensis* and *L. amazonensis* promastigotes (see Table 1). Their antiparasitic activities were compared with those previously reported for the analogues in the **1**-series in addition to compounds **2a** (pulchral), **3a**, **4a**, and **5a** [44]. Figure 2 summarizes the structure types of the analogues prepared, while the biological activities are given in Table 1. 

Table 2 and Table 3 give the 1D ^1^H and ^13^C NMR shifts of the assayed compounds. 

### 2.2. Selected Functionalities 

The natural products pulchrol (**1a**) and pulchral (**2a**) have been studied in the past, and both of them have been shown to be active against *Trypanosoma* and *Leishmania* parasites [44]. Pulchrol (**1a**) has been reported to be moderately active against *L. braziliensis* and *L. amazonensis* promastigotes (IC_50_ 59.2 μM and IC_50_ 77.7 μM, respectively), while it has been shown to possess potent toxicity towards *T. cruzi* epimastigotes (IC_50_ 18.5 μM), comparable to that of the drug benznidazole (19.2 μM) that currently is used as treatment for Chagas disease. Meanwhile, pulchral (**2a**) has been shown to be active against all of the three parasites (*T. cruzi*, IC_50_ 24.2 μM; *L. braziliensis*, IC_50_ 24.2 μM; *L. amazonensis*, IC_50_ 29.8 μM) [44,45].

In previous investigations, we studied the effect that individual transformations in the A-, B-, and C-rings have on the activity towards *T. cruzi*, *L. braziliensis* and *L. amazonensis* [44,45]. Initially, we reported the effect of the modifications at position 1′ in the A-ring, and concluded preliminarily that the benzyl alcohol functionality was important for pulchrol’s activity towards all of the three parasites, possibly acting as a hydrogen bond acceptor. It was also observed that the 1′-carbonyl analogues were equipotent towards *T. cruzi* compared to pulchrol, while they were more potent towards *L. braziliensis* and *L. amazonensis*. On the other hand, the 9-methyl analogue was reported to possess considerably less activity than pulchrol against all the parasites. Finally, the ester analogues were shown to be, in general, more potent than pulchrol, with longer and branched alkyl substituents showing considerable improvements [44]. The effects that modifications in the B- and C-rings have on the antiparasitic activity were also investigated, and were mainly focused on the effects that variations in the lipophilicity may have in the antiparasitic activity. The 6-monoalkyl (methyl and ethyl) derivatives, also as pure enantiomers, were less potent towards the parasites compared to pulchrol, and a preference for 6,6-dialkyl analogues was established. Alkyl substituents in the C-ring were shown to be beneficial for the antiparasitic activity; most notably, longer and branched alkyl substituents at positions 2 and 3 increased the potency considerably [45].

## 3. Discussion

In this study, we analyse the effect that two or three transformations in the different rings of the pulchrol scaffold may have on the antiparasitic activity. Most of the analogues have transformations in the A- and C-rings, whereas just a pair have the A- and B-rings modified. The 1′-aldehyde, the 3-methyl butanoic acid ester, and the 9-methyl functionalities in the A-ring were combined with isopropyl substituents at C-2 and C-3 in the C-ring (**2b** and **2c**; **4b** and **4c**; and **5b** and **5c,** respectively) to see how an increase in lipophilicity would affect their antiparasitic activity. An analogue with the 1′-methyl ketone functionality in the A-ring and an isopropyl substituent at position 2 was also prepared (**3b**) to see if any differences would arise with respect to the corresponding 1′-aldehyde analogue (**2b**). The 9-methyl functionality was also combined with a hydroxyl substituent at C-2 in the C-ring (**5d**), and the natural compound cannabinol (**5f**) and its analogue **5e** were prepared, both of them containing a 9-methyl substituent in the A-ring, a hydroxyl group at C-1, and an alkyl group at C-3 in the C-ring; cannabinol (**5f**) with an *n*-pentyl substituent and **5e** with a methyl substituent at C-3. Finally, the 1′-aldehyde functionality was combined with the 6-monomethyl substituent in the B-ring, as a pair of enantiomers (**2d** and **2e**), to evaluate whether the carbonyl group helps to improve their activity or hamper it in comparison with the already reported benzyl alcohol enantiomers **1d** and **1e** [45].

### 3.1. Antiparasitic Activities towards Trypanosoma Cruzi Epimastigotes

It has previously been reported that the transformation of the benzyl alcohol functionality in pulchrol (**1a,** IC_50_ = 18.5 μM) to a 1′-carbonyl group (**2a**, IC_50_ = 24.2 μM and **3a**, IC_50_ = 21.3 μM) was not beneficial for the antiparasitic activity towards *T. cruzi* [44], while the replacement of the methoxy functionality (in **1a**) for an isopropyl group in positions 2 and 3 in the C-ring was shown to improve the IC_50_ values for this parasite (**1b**, IC_50_ = 12.4 μM and **1c**, IC_50_ = 14.2 μM, respectively) [45]. In order to evaluate the effect that combinations involving the functionalities mentioned before may have in the antitrypanosomal activity, analogues with an aldehyde functionality in the A-ring and with an isopropyl substituent at positions 2 or 3 in the C-ring were prepared and found to be considerably more active (**2b**, IC_50_ = 10.7 μM; and **2c**, IC_50_ = 7.1 μM) compared to the previously reported corresponding benzyl alcohols **1b** and **1c** [45]. This could be caused by a change in orientation in a hypothetical binding site, by the combined functionalities, possibly enhancing the hydrophobic interactions of the isopropyl substituents in **2b** and **2c**. The methyl ketone **3b**, substituted with an isopropyl group at C-2, was actually the most potent compound towards *T. cruzi* (IC_50_ = 3.4 μM) of all the analogues prepared in this study, showing six-fold higher activity than the positive control benznidazole (IC_50_ = 19.2 μM), indicating possible interactions between the methyl substituent in the ketone and the active site. 

Contrary to the improvements shown by the analogues substituted with an isopropyl group in the **2**- and **3**-series, 2- and 3-isopropyl analogues of the **4**-series (combined with a 3-methylbutanoic acid ester) were less potent towards *T. cruzi* (**4b**, IC_50_ = 10.9 μM; **4c**, IC_50_ = 13.6 μM) compared to the corresponding 2-methoxy ester **4a** (IC_50_ = 4.2 μM), which has previously been reported to possess better activity than the positive control benznidazole [44]. 

Among the analogues of the **5**-series, which contain a 9-methyl substituent in the A-ring, analogue **5a** (with a methoxy group at C-2, IC_50_ = 51.1 μM) has previously been shown to possess less potency than pulchrol (**1a**) [44]. Similarly, in this study analogue **5d,** with a hydroxyl functionality at C-2, showed comparable activity (IC_50_ = 54.8 μM) to **5a** towards *T. cruzi*. It is possible that both analogues, **5a** and **5d,** interact with the binding site in a rotated position, in which the methoxy group in **5a** and the hydroxy group in **5d** act as hydrogen bond acceptors where pulchrol’s benzyl alcohol usually interacts. On the other hand, more lipophilic analogues, **5b** and **5c**, with isopropyl substituents at positions 2 and 3 (IC_50_ = 23.7 μM and IC_50_ = 50.3 μM, respectively) were found to be less toxic than pulchrol (**1a**, IC_50_ = 18.5 μM)**.** However**,** cannabinol (**5f**) and its analogue **5e**, which also has an alkyl substituent at C-3 and, in addition, a hydroxyl functionality at C-1, were shown to be the most potent (**5f**, IC_50_ = 7.4 μM; **5e**, IC_50_ = 5.9 μM) compounds from the **5**-series towards *T. cruzi*. Analogue **5e,** with a methyl substituent at C-3 instead of the *n*-pentyl group in cannabinol (**5f**), is slightly more active than **5f** and possesses one of the highest activities towards *T. cruzi* among all the derivatives prepared in this project, possibly due to interactions between the hydroxyl group at C-1 and the target protein. 

Finally, the aldehyde enantiomers **2d** and **2e**, substituted with a single methyl group at position 6 in the B-ring, possess considerably lower antiparasitic activity compared to pulchrol (**1a**, IC_50_ = 18.5 μM). They are also less potent compared to their synthetic precursors, the benzyl alcohols **1d** and **1e** [45], indicating that a possible change in orientation may disrupt hydrophobic interactions around position 6 in the B-ring. 

### 3.2. Antiparasitic Activities towards Leishmania Braziliensis Promastigotes

The opposite to what has been reported for *T. cruzi*, the transformation of pulchrol’s benzyl alcohol to a 1′-carbonyl functionality (**2a** and **3a**) has been shown to considerably increase the antiparasitic activity towards *L. braziliensis* (**2a**, IC_50_ = 24.2 μM; and **3a**, IC_50_ = 28.3 μM) [44]. Likewise, isopropyl substituents in the C-ring (**1b** and **1c**) also increase the potency (**1b**, IC_50_ = 18.1 μM; and **3a**, IC_50_ = 19.1 μM) compared to pulchrol (**1a**, IC_50_ = 59.2 μM) [45]. In this study, the aldehydes **2b** and **2c** (with isopropyl substituents at C-2 and C-3, respectively**)** were prepared and found to be more toxic towards *L. braziliensis* (**2b**, IC_50_ = 12.1 μM; and **2c**, IC_50_ = 17.8 μM) than their benzyl alcohol precursors (**1b** and **1c**). As for *T. cruzi*, the 2-isopropyl ketone **3b** was the most potent compound (IC_50_ = 8.8 μM) in this investigation. Furthermore, the 1′-carbonyl analogues **2b** and **3b** are also more potent than the positive control miltefosine (IC_50_ = 13.0 μM). Possibly, the increase in potency shown by the carbonyl analogues combined with alkyl substituents in the C-ring, may be due to better hydrophobic interactions with a target protein, produced by a change in the orientation of the alkyl groups. 

Similar results to those obtained for *T. cruzi* were observed for the analogues from the **4**-series; the 2- and 3-isopropyl esters **4b** and **4c** were less active towards *L. braziliensis* (**4b**, IC_50_ = 272.9 μM; and **4c**, IC_50_ = 63.3 μM) than the previously reported ester **4a** (with a methoxy substituent at C-2, IC_50_ = 13.1 μM) [44]. The opposite to *T. cruzi*, analogue **4b** was inactive, indicating a possible limitation in the volume of the lipophilic pocket where the 3-alkyl substituents interact.

The transformation of the benzyl alcohol moiety in pulchrol (**1a**) to a 9-methyl substituent, as in **5a**, has previously been determined to be unfavourable for the activity (**1a**, IC_50_ = 59.2 μM; and **5a**, IC_50_ = 69.6 μM) [44]. In this investigation, the 9-methyl/2-isopropyl analogue **5b** and the 9-methyl/2-hydroxy analogue **5d** are more potent towards *L. braziliensis* (**5b**, IC_50_ = 49.2 μM; and **5d**, IC_50_ = 30.4 μM), opposite to *T. cruzi*. On the other hand, the 9-methyl/3-isopropyl analogue **5c** was shown to be essentially inactive (IC_50_ = 312 μM), suggesting again that there is a limit in the volume around positions 2 and 3. Similar to the results obtained for *T. cruzi*, **5e** and cannabinol (**5f**) were found to possess higher activity (IC_50_ = 15.7 μM and 10.3 μM, respectively) than **5a** and pulchrol (**1a**). However, the longer chain at C-3 in cannabinol (**5f**) seems to be better for the activity against *L. braziliensis* compared to **5e**, with a methyl group at C-3, while the opposite is true for *T. cruzi*.

As with *T. cruzi*, the aldehyde enantiomers **2d** and **2e** were found to be less potent (IC_50_ = 70.8 μM and 118 μM, respectively) compared to pulchrol (**1a**), although they are more potent than their corresponding benzyl alcohol enantiomers **1e** and **1f** (IC_50_ = 156 μM and 129 μM, respectively) [45]. 

### 3.3. Antiparasitic Activities towards Leishmania Amazonensis Promastigotes

For *L. amazonensis*, as with *L. braziliensis*, pulchral (**2a**), and the analogues **1b** and **1c** (substituted with an isopropyl group at C-2 and C-3 in the C-ring, respectively) have been reported to be more potent than pulchrol [44,45]. Similar to *T. cruzi* and *L. braziliensis*, the combination of the 9-aldehyde functionality with isopropyl substituents at positions 2 and 3 (**2b** and **2c**, respectively) is beneficial for the antiparasitic activity, and the highest activities were shown by the aldehyde **2b** (IC_50_ = 11.4 μM), and the methyl ketone **3b** (IC_50_ = 9.5 μM), which are equipotent with the positive control miltefosine (IC_50_ = 10.8 μM). All the ester analogues (**4a** to **4b**) were more toxic than pulchrol towards *L. amazonensis*, and as with the other parasites, the previously reported **4a** (with a methoxy substituent on C-2) was still the most potent (IC_50_ = 14.5 μM) among the esters.

The analogues possessing a 9-methyl substituent were similarly as potent towards *L. amazonensis* as towards *L. braziliensis*. Most of the 9-methyl analogues (**5b**, **5d**, **5e** and **5f**) are more potent than pulchrol (**1a**, IC_50_ = 77.7 μM), while the 9-methyl/3-isopropyl analogue **5c** is much less active (IC_50_ = 236.5 μM) towards *L. braziliensis*. The most potent compounds in the **5**-series are cannabinol (**5f**, IC_50_ = 14.2 μM) and its analogue **5e** (IC_50_ = 21.2 μM). Finally, the enantiomers **2d** (IC_50_ = 44.0 μM) and **2e** (IC_50_ = 80.6 μM) were still less potent than the corresponding 6,6-dimethyl aldehyde **2a** (pulchral, IC_50_ = 29.8 μM) [44], similar to what was observed for *L. braziliensis* and *T. cruzi*. 

## 4. Materials and Methods

### 4.1. General

^1^H NMR spectra (400 MHz) and ^13^C NMR spectra (100 MHz) (See Appendix A) were recorded with a Bruker Avance II (Bruker Biospin AG, Industriestrasse 26, 8117 Fällanden, Switzerland) in CDCl_3_. The individual 1D signals were assigned using 2D NMR experiments (COSY, HSQC, HMBC). The chemical shifts are given in ppm with the solvent signal as reference (7.27 ppm for ^1^H and 77.0 for ^13^C). Infrared spectra were recorded with a Bruker Alpha-P FT/IR instrument (Bruker Biospin AG, Industriestrasse 26, 8117 Fällanden, Switzerland) with a Diamond ATR sensor as films, and the intensities are given as vw (very weak), w (weak), m (medium), s (strong) and vs (very strong). High-resolution mass spectra (HRMS) were recorded with Waters XEVO-G2 QTOF equipment (Waters Corp, Milford, Worcester County, MA, USA), with electrospray ionization (ESI). Synthetic reactions were monitored by TLC using alumina plates coated with silica gel and visualized using either UV light and/or spraying/heating with vanillin/H_2_SO_4_. Flash chromatography was performed with silica gel (35–70 μm, 60 Å). THF was distilled from sodium, acetonitrile was distilled from CaH_2_ and other reaction solvents were dried with Al_2_O_3_. Commercially available compounds were obtained from Aldrich.

### 4.2. Synthetic Procedures

*Methyl 4-(hydroxymethyl)-2-iodobenzoate* (intermediate in the synthesis of **1a**–**1e**): BH_3_-THF (1 M, 47.1 mL, 47.1 mmol) was slowly added to a stirred solution of 1-methyl-2-iodoterephthalate (4.8 g, 15.7 mmol) in dry THF (250 mL) at 0 °C. After 30 h, saturated aqueous NaHCO_3_/H_2_O was added, and the aqueous phase was extracted with ethyl acetate (3 × 250 mL) before drying (Na_2_SO_4_) and removal of solvent under reduced pressure. Purification by column chromatography (SiO_2_, 4:6 heptane/ethyl acetate) gave 4.06 g (89%) of the pure product as yellow crystals, identical to that previously reported [42].

*Methyl 4-(((tert-butyldiphenylsilyl)oxy)methyl)-2-iodobenzoate* (intermediate in the synthesis of **1a**–**1e**): TBDPSCl (4.3 mL, 16.7 mmol) was added to a stirred solution of methyl 4-(hydroxymethyl)-2-iodobenzoate (4.06 g, 13.9 mmol) in pyridine (80 mL) at rt. After 24 h, saturated aqueous NH_4_Cl/H_2_O was added and the aqueous phase was extracted with diethyl ether (3 × 200 mL), then the organic phase was washed with brine (2 × 500 mL) before drying (Na_2_SO_4_) and removal of solvent under reduced pressure. Purification by column chromatography (SiO_2_, 20:2 heptane/ethyl acetate) gave 4.2 g (57%) of the pure product as white crystals, identical to that previously reported [42].

*General procedure for Suzuki coupling* (intermediates in the synthesis of **1a**–**1e**): corresponding boronic acid (1.5 equiv), K_2_CO_3_ (5 equiv) and Tetrakis(triphenylphosphine)palladium(0) (0.17 equiv) were added to a stirred solution of methyl 4-(((tert-butyldiphenylsilyl)oxy)methyl)-2-iodobenzoate (1 equiv) dissolved in 4:1 DME/water (15 mL), the mixture (contained in a microtube) was degasified under vaccuum/N_2_ at −78 °C five times. The microwave reaction conditions were 100 °C, high pressure, and 10 s of pre-stirring. After 30 to 60 min in the microwave reactor, the mixture was filtered through a plug of celite and washed with ethyl acetate (250 mL) before drying (Na_2_SO_4_) and removal of solvent under reduced pressure. Purification by column chromatography (SiO_2_, 20:3 heptane/ethyl acetate) gave the pure products.

*Methyl 5-(((tert-butyldiphenylsilyl)oxy)methyl)-2′,5′-dimethoxy-[1,1′-biphenyl]-2-carboxylate* (intermediate in the synthesis of **1a**): the pure product was obtained as an orange wax (yield 94%) identical to that previously reported [42].

*Methyl 5-(((tert-butyldiphenylsilyl)oxy)methyl)-5′-isopropyl-2′-methoxy-[1,1′-biphenyl]-2-carboxylate* (intermediate in the synthesis of **1b**): the pure product was obtained as colorless wax (yield 97%) identical to that previously reported [45].

*Methyl 5-(((tert-butyldiphenylsilyl)oxy)methyl)-2′-methoxy-4′-isopropyl-[1,1′-biphenyl]-2-carboxylate* (intermediate in the synthesis of **1c**): the pure product was obtained as a yellowish wax (yield 38%) identical to that previously reported [45]. 

*General procedure for organo-lithic addition* (intermediate in the synthesis of **1a**–**1e**): Corresponding organo-lithic reagent (4 equiv) was added to a stirred solution of the Suzuki coupling product (1 equiv) in dry THF (70 mL), at 0 or −78 °C, depending on the organo-lithic reagent. After 12 h, saturated aqueous NH_4_Cl/H_2_O was added, and the aqueous phase was extracted with ethyl acetate (3 × 100 mL) before drying (Na_2_SO_4_) and removal of solvent under reduced pressure. Purification by column chromatography (SiO_2_, 20:4 heptane/ethyl acetate) gave the pure product.

*2-(5-(((tert-butyldiphenylsilyl)oxy)methyl)-2′,5′-dimethoxy-[1,1′-biphenyl]-2-yl)propan-2-ol* (intermediate in the synthesis of **1a**): the pure product was obtained as a yellowish wax (yield 65.2%) identical to that previously reported [42].

*2-(5-(((tert-butyldiphenylsilyl)oxy)methyl)-5′-isopropyl-2′-methoxy-[1,1′-biphenyl]-2-yl)propan-2-ol* (intermediate in the synthesis of **1b**): the pure product was obtained as a transparent wax (yield 78.5%) identical to that previously reported [45].

*2-(5-(((tert-butyldiphenylsilyl)oxy)methyl)-2′-methoxy-4′-isopropyl-[1,1′-biphenyl]-2-yl)propan-2-ol* (intermediate in the synthesis of **1c**): the pure product was obtained as a transparent wax (yield 64%) identical to that previously reported [45].

*General procedure to prepare compounds* **1a**–**1c**: HI (55%, 10 equiv) was added to a stirred solution of the corresponding starting material in acetonitrile (25 mL), at rt. After 30 min, saturated aqueous Na_2_S_2_O_3_ (25 mL) was added, and the aqueous layer was extracted with ethyl acetate (3 × 50 mL), before drying (Na_2_SO_4_) and removal of solvent under reduced pressure. TBAF (1 M, 1.1 equiv) was added to the crude product in THF (150 mL). After 3 h, aqueous saturated NaHCO_3_ (50 mL) was added, and the aqueous layer was extracted with ethyl acetate (3 × 50 mL), before drying (Na_2_SO_4_) and removal of solvent under reduced pressure. Purification by column chromatography (SiO_2_, 1:1 heptane/ethyl acetate) gave the pure product.

*(2-methoxy-6,6-dimethyl-6H-benzo[c]chromen-9-yl)methanol* (**1a**): the pure product was obtained as a yellowish wax (yield 77%) identical to that previously reported [42].

*(2-isopropyl-6,6-dimethyl-6H-benzo[c]chromen-9-yl)methanol* (**1b**): the pure product was obtained as a transparent wax (yield 88%) identical to that previously reported [45].

*(3-isopropyl-6,6-dimethyl-6H-benzo[c]chromen-9-yl)methanol* (**1c**): the pure product was obtained as a transparent wax (yield 85%) identical to that previously reported [45].

*5-(((tert-butyldiphenylsilyl)oxy)methyl)-2′,5′-dimethoxy-[1,1′-biphenyl]-2-carbaldehyde* (intermediate in the synthesis of **1d** and **1e**): Morpholine (0.2 mL, 2.2 mmol) was added to a solution of DIBALH (1 M, 1.1 mL, 1.1 mmol) in dry THF (30 mL) at 0 °C. After 3 h, methyl 5-(((*tert*-butyldiphenylsilyl)oxy)methyl)-2′,5′-dimethoxy-[1,1′-biphenyl]-2-carboxylate (600 mg, 1.1 mmol) in dry THF (20 mL) was added, 10 min later, DIBALH (1 M, 1.1 mL, 1.1 mmol) was added again at 0 °C. After 4 h, aqueous HCL (1 N, 20 mL) was added, and the aqueous phase was extracted with diethyl ether (3 × 50 mL) before drying (Na_2_SO_4_) and removal of solvent under reduced pressure. Purification by column chromatography (SiO_2_, 20:4 heptane/ethyl acetate) gave the pure product as a yellowish wax (89.1 mg, 16%) identical to that previously reported [45].

*General procedure to prepare compounds* **1d** and **1e**: MeLi (3 M, 2 equiv) was added to 5-(((tert-butyldiphenylsilyl)oxy)methyl)-2′,5′-dimethoxy-[1,1′-biphenyl]-2-carbaldehyde (1 equiv) in dry THF (5 mL) at 0 °C. After 6 h, saturated aqueous NH_4_Cl/H_2_O was added, and the aqueous phase was extracted with ethyl acetate (3 × 20 mL) before drying (Na_2_SO_4_) and removal of solvent under reduced pressure. PBr_3_ (0.34 equiv) was added to the crude product (1 equiv) in dichloromethane (10 mL) at rt. After 2 h, LiI (3 equiv) was added at rt. After 12 h, saturated aqueous Na_2_S_2_O_3_/H_2_O was added, and the aqueous phase was extracted with ethyl acetate (3 × 20 mL) before drying (Na_2_SO_4_) and removal of solvent under reduced pressure. TBAF (2 equiv) was added to the crude product in THF (25 mL) at rt. After 5 h, saturated aqueous NaHCO_3_/H_2_O was added, and the aqueous phase was extracted with ethyl acetate (3 × 25 mL) before drying (Na_2_SO_4_) and removal of solvent under reduced pressure. Purification by column chromatography (SiO_2_, 1:1 heptane/ethyl acetate) and the enantiomers were separated using a semipreparative HPLC (Chiralpack B column, 96:4 hexane/isopropanol).

*(2-methoxy-6-methyl-6H-benzo[c]chromen-9-yl)methanol* (**1d**): the pure product was obtained as a yellowish wax (yield 3%), αD20-17.8 identical to that previously reported [45].

*(2-methoxy-6-methyl-6H-benzo[c]chromen-9-yl)methanol* (**1e**): the pure product was obtained as a yellowish wax (yield 3%), αD20 +23.8° identical to that previously reported [45].

*General procedure to obtain compounds* **2b***–***2e**: Dess–Martin periodinane 15% (2 equiv) was added to a stirred solution of corresponding starting material (1 equiv) in CH_2_Cl_2_ at rt. After five hours, saturated aqueous Na_2_S_2_O_3_/H_2_O was added and the aqueous phase was extracted three times with CH_2_Cl_2_ before drying (Na_2_SO_4_) and removal of solvent under reduced pressure. Purification by column chromatography (SiO_2_, 20:1 CH_2_Cl_2_/methanol). 

*2-isopropyl-6,6-dimethyl-6H-benzo[c]chromene-9-carbaldehyde* (**2b**): The pure product was obtained as a transparent wax (85.7 mg, yield 88%). ^1^H NMR and ^13^C NMR data in Table 2 and Table 3, respectively. HRMS-ESI+ (*m/z*): [M + Na]^+^ calcd for C_19_H_20_O_2_, 303.1361; found 303.1355.

*3-isopropyl-6,6-dimethyl-6H-benzo[c]chromene-9-carbaldehyde* (**2c**): the pure product was obtained as a transparent wax (15.3 mg, yield 77%). ^1^H NMR and ^13^C NMR data in Table 2 and Table 3, respectively. HRMS-ESI+ (*m/z*): [M + Na]^+^ calcd for C_19_H_20_O_2_, 303.1361; found 303.1357.

*2-methoxy-6-methyl-6H-benzo[c]chromene-9-carbaldehyde* (**2d**): the pure product was obtained as a yellowish wax (5.4 mg, yield 96%). ^1^H NMR and ^13^C NMR data in Table 2 and Table 3, respectively. HRMS-ESI+ (*m/z*): [M + Na]^+^ calcd for C_16_H_14_O_5_, 309.0738; found 309.0739 (+2O due to oxidation).

*2-methoxy-6-methyl-6H-benzo[c]chromene-9-carbaldehyde* (**2e**): the pure product was obtained as a yellowish wax (6.7 mg, yield 95%). ^1^H NMR and ^13^C NMR data in Table 2 and Table 3, respectively. HRMS-ESI+ (*m/z*): [M + Na]^+^ calcd for C_16_H_14_O_5_, 309.0739; found 309.0739 (+2O due to oxidation).

*1-(2-isopropyl-6,6-dimethyl-6H-benzo[c]chromen-9-yl)ethan-1-one* (**3b**): MeMgI (3 M, 0.24 mL, 0.7 mmol) was added to **2b** (65 mg, 0.23 mol) in dry ethyl ether (5 mL), at 0 °C. After 20 h, saturated aqueous NH_4_Cl/H_2_O was added, and the aqueous phase was extracted with diethyl ether (3 × 5 mL). The organic product was washed with brine once before drying (Na_2_SO_4_) and removal of solvent under reduced pressure. Celite (250 mg) and PCC (63 mg, 0.25 mmol) were added to the crude product in dry CH_2_Cl_2_ (5 mL) under nitrogen, at rt. After 12 h diethyl ether (10 mL) was added and the mixture was filtered over a plug of silica gel and washed with ethyl acetate (50 mL) before removal of solvent under reduce pressure. Purification by column chromatography (SiO_2_, 20:4 hept/ethyl acetate) gave 30.0 mg, 44% as a transparent wax.%). ^1^H NMR and ^13^C NMR data in Table 2 and Table 3, respectively. HRMS-ESI+ (*m/z*): [M + Na]^+^ calcd for C_20_H_22_O_2_, 317.1514; found 317.1517.

*General procedure to obtain compounds* **4b**–**4c**: Isovaleric anhydride (1.5 equiv), DMAP (1.2 equiv), and Et_3_N (1.5 equiv) were added to a stirred solution of appropriate starting material (1 equiv) in CH_2_Cl_2_ (25 mL) at rt. After three hours, saturated aqueous NH_4_Cl/H_2_O was added and the aqueous phase was extracted with CH_2_Cl_2_ (3 × 25 mL) before drying (Na_2_SO_4_) and removal of solvent under reduced pressure. Purification by column chromatography (SiO_2_, 20:4 heptane/ethyl acetate) gave the desired products.

*(2-isopropyl-6,6-dimethyl-6H-benzo[c]chromen-9-yl)methyl 3-methylbutanoate* (**4b**): The pure product was obtained as a transparent wax (64.9, yield 53%). ^1^H NMR and ^13^C NMR data in Table 2 and Table 3, respectively. HRMS-ESI+ (*m/z*): [M + Na]^+^ calcd for C_24_H_30_O_3_, 389.2093; found 389.2099.

*(3-isopropyl-6,6-dimethyl-6H-benzo[c]chromen-9-yl)methyl 3-methylbutanoate* (**4c**): The pure product was obtained as a transparent wax (15.2, yield 59%). ^1^H NMR and ^13^C NMR data in Table 2 and Table 3, respectively. HRMS-ESI+ (*m/z*): [M + Na]^+^ calcd for C_24_H_30_O_3_, 389.2093; found 389.2091.

*General procedure to obtain compounds* **5a**–**5c**: Et_3_SiH (5 equiv) and PdCl_2_ (2 equiv) were added to a stirred solution of corresponding starting material (1 equiv) in EtOH. After 3h, ethyl acetate was added and the mixture was filtered through a plug of celite and washed with ethyl acetate before drying (Na_2_SO_4_) and removal of solvent under reduced pressure. Purification by column chromatography (SiO_2_, 20:1 heptane/ethyl acetate). 

*2-Methoxy-6,6,9-trimethyl-6H-benzo[c]chromene* (**5a**): the pure product was obtained as a yellowish wax (yield 92%) identical to that previously reported [44].

*2-isopropyl-6,6,9-trimethyl-6H-benzo[c]chromene* (**5b**): the pure product was obtained as a transparent wax (105 mg, yield 89%). ^1^H NMR and ^13^C NMR data in Table 2 and Table 3, respectively. HRMS-ESI+ (*m/z*): [M + Na]^+^ calcd for C_19_H_22_O, 289.1568; found 289.1563.

*3-isopropyl-6,6,9-trimethyl-6H-benzo[c]chromene* (**5c**): the pure product was obtained as a transparent wax (16 mg, yield 85%). ^1^H NMR and ^13^C NMR data in Table 2 and Table 3, respectively. HRMS-ESI+ (*m/z*): [M + Na]^+^ calcd for C_19_H_22_O, 289.1568; found 289.1565.

*6,6,9-trimethyl-6H-benzo[c]chromen-2-ol* (**5d**): Sodium ethanethiolate (132 mg, 1.57 mmol2) was added to a stirred solution of **5a** (0.157 mmol) in DMF (4 mL), and reacted in a microwave reactor at 160 °C. After one hour, saturated aqueous NH_4_Cl/H_2_O was added and the aqueous phase was extracted with ethyl acetate (3 × 100 mL) before drying (Na_2_SO_4_) and removal of solvent under reduced pressure. Purification by column chromatography (SiO_2_, 20:4 heptane/ethyl acetate) gave 27 mg (72%) of the pure product as an orange wax.%). ^1^H NMR and ^13^C NMR data in Table 2 and Table 3, respectively. HRMS-ESI+ (*m/z*): [M + H]^+^ calcd for C_16_H_17_O_2_, 241.1229; found 241.1231.

*General procedure to obtain compounds* **5e***and* **5f**: Citral (1 equiv) and *n*-butylamine (1 equiv) were added to the corresponding resorcinol analogue (1 equiv) in toluene at 110 °C. After 12 h, DOWEX 50WX8 (200 mg) was added to the stirred mixture at rt. After 10 min the mixture was filtered over a plug of celite. Iodine (2 equiv) was added to the filtrate at 110 °C. After 3 h, saturated aqueous Na_2_S_2_O_3_/H_2_O was added and the aqueous phase was extracted three times with EtOAc before drying (Na_2_SO_4_) and removal of solvent under reduced pressure. Purification by column chromatography (SiO_2_, 100:5 heptane/ethyl acetate).

*3,6,6,9-tetramethyl-6H-benzo[c]chromen-1-ol* (**5e**): the pure product was obtained as an orange wax (26 mg, yield 18%). ^1^H NMR and ^13^C NMR data in Table 2 and Table 3, respectively. HRMS-ESI+ (*m/z*): [M + Na]^+^ calcd for C_17_H_18_O_2_, 277.1204; found 277.1201.

*6,6,9-trimethyl-3-pentyl-6H-benzo[c]chromen-1-ol* (**5f**): the pure product was obtained as a brownish wax (6.7 mg, yield 4%). ^1^H NMR and ^13^C NMR data in Table 2 and Table 3, respectively. HRMS-ESI+ (*m/z*): [M + Na]^+^ calcd for C_21_H_26_O_2_, 333.1830; found 333.1825.

### 4.3. Biological Assays

#### 4.3.1. Evaluations against *Leishmania* Parasites 

Promastigotes of Leishmania (Leishmania): *L. amazonensis*, clone 1, NHOM-BR-76-LTB-012 (Lma, donated by the Paul Sabatier Université, France) and Leishmania (Viannia): *L. braziliensis* M2904 C192 RJA (M2904, donated by Dr. Jorge Arévalo from Universidad Peruana Cayetano Heredia, Peru) [47]. All strains were cultured in Schneider’s insect medium (pH 6.2), supplemented with 10% FBS and incubated at 26 °C. Medium changes were made every 72 h to maintain a viable parasitic population. Leishmanicidal activity was determined according to Williams with some modifications [48]. Samples were dissolved in DMSO (maximum final concentration 1%) at 10 mg/mL. Promastigotes in logarithmic phase of growth, at the concentration 3 × 10^6^ parasites/mL, were distributed (100 μL/well) in 96-well flat bottom microtiter plates. Samples with different concentrations (3.1–100 μg/mL) were added (100 μL). Miltefosine (3.1–100 μg/mL) was used as control drug [49]. Assays were performed in triplicates. The microwell plates were incubated for 72 h at 26 °C. After incubation, a solution of XTT (1 mg/mL, [2,3-bis[2-methoxy-4-nitro-5-sulfophenyl]-2H-tetrazolium-5-carboxanilide]) in PBS (pH 7.0 at 37 °C) with PMS (0.06 mg/mL phenazine methosulfate) was added (50 μL/well), and incubated for 3 h at 26 °C. All assays were carried out as triplicates. The optical density of each well was measured with a Synergy HT microplate reader at 200–450 nm. The IC_50_ values were calculated using the Gen5 program (BioTek).

#### 4.3.2. Evaluations against *Trypanosoma cruzi*

Cultures of *Trypanosoma cruzi* (epimastigotes, donated by the Parasitology Department of INLASA, Tc-INLASA) were maintained in medium LIT (liver infusion tryptose) (pH 7.2), supplemented with 10% FBS and incubated at 26 °C. The LIT medium was prepared with NaCl, 40 g/L; KCl, 4 g/L; Na_2_HPO_4_, 80 g/L; tryptose, 50 g/L; 10% liver infusion broth, 50 mL/L; 40% glucose solution, 10 mL/L; penicillin, 200 units/mL; streptomycin, 200 units/mL; and neomycin, 200 units/mL. Medium changes were made every 72 h to maintain a viable parasitic population. Trypanocidal activity was determined according to Muelas-Serrano with some modifications [50]. Samples were dissolved in DMSO (maximum final concentration 1%) at 10 mg/mL. Epimastigotes in logarithmic phase of growth, at a concentration of 3 × 10^6^ parasites/mL, were distributed (100 μL/well) in 96-well flat bottom microtiter plates. Samples at different concentrations (3.1–100 μg/mL) were added (100 μL). Benznidazol (3.1–100 μg/mL) was used as the control drug. Assays were performed in triplicates. The microwell plates were incubated for 72 h at 26 °C. After incubation, a solution of XTT (1 mg/mL) in PBS (pH 7.0 at 37 °C) with PMS (0.06 mg/mL) was added (50 μL/well) and incubated for 4 h at 26 °C. All assays were carried out as triplicates. The optical density of each well was measured with a Synergy HT microplate reader at 200–450 nm. The IC_50_ values were calculated using the Gen5 program (BioTek).

## 5. Conclusions

Twelve compounds with chemical transformations of the benzyl alcohol functionality in the A-ring, and with modifications either in the C-ring or the B-ring, were prepared and assayed towards *T. cruzi*, *L. braziliensis* and *L. amazonensis*. Although *Trypanosoma* and *Leishmania* species belong to the same order (Trypanosomatida), they apparently possess different molecular targets in their different life stages, and the ways they infect humans are not fully understood. Moreover, the active site or active sites on which pulchrol acts on each of the parasites are unknown; therefore, the bioactivity results reported in this work just provide some suggestions of SARs and the surrounding near a possible binding site. The effects of the antiparasitic activity that lipophilic groups in the C-ring may have was studied by testing analogues substituted with isopropyl groups at position 2 and 3, combined with different transformations of pulchrol’s benzyl alcohol functionality. The combination of a 1′-carbonyl group in the A-ring was found to be beneficial for the activity towards all of the three parasites; however, the 3-isopropyl analogues showed slightly better leishmanicidal activity than the 2-isopropyl analogues. The latter are more potent towards *T. cruzi*, suggesting that differences in the binding sites of the protein targets may exist. The most interesting antiparasitic activity was showed by the analogue containing a 1′-methyl ketone functionality in the A-ring and the isopropyl group at position 2. Moreover, all the carbonyl/isopropyl analogues were more potent than the analogues containing just the carbonyl functionality in the A-ring or just the isopropyl group in the C-ring, indicating that the combination may improve the orientation in the binding site.

## Figures and Tables

**Figure 1 molecules-26-03944-f001:**
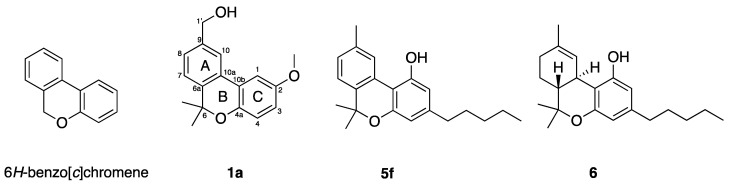
The 6*H*-benzo[*c*]chromene scaffold studied here (**left**), and structures of pulchrol (**1a**), cannabinol (**5f**) and Δ^9^-tetrahydrocannabinol (**6**).

**Figure 2 molecules-26-03944-f002:**
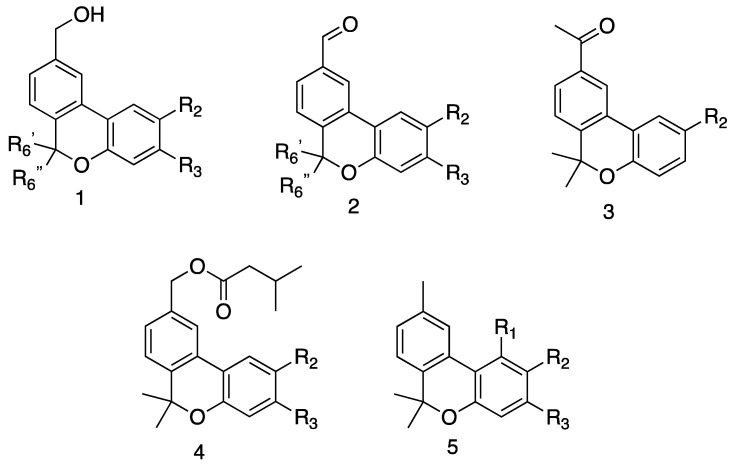
Prepared derivatives. **1a** R_2_ = OMe, R_3_ = H, R_6_′ = Me, R_6_″ = Me; **1b** R_2_ = isopropyl, R_3_ = H, R_6_′ = Me, R_6_″ = Me; **1c** R_2_ = H, R_3_ = isopropyl, R_6_′ = Me, R_6_″ = Me; **1d** R_2_ = OMe, R_3_ = H, R_6_′ = Me, R_6_″ = H; **1e** R_2_ = OMe, R_3_ = H, R_6_′ = H, R_6_″ = Me; **2a** R_2_ = OMe, R_3_ = H, R_6_′ = Me, R_6_″ = Me; **2b** R_2_ = isopropyl, R_3_ = H, R_6_′ = Me, R_6_″ = Me; **2c** R_2_ = H, R_3_ = isopropyl, R_6_′ = Me, R_6_″ = Me**; 2d** R_2_ = OMe, R_3_ = H, R_6_′ = Me, R_6_″ = H; **2e** R_2_ = OMe, R_3_ = H, R_6_′ = H, R_6_″ = Me; 3**a** R_2_ = OMe; **3b** R_2_ = isopropyl; **4a** R_2_ = OMe, R_3_ = H; **4b** R_2_ = isopropyl, R_3_ = H; **4c** R_2_ = H, R_3_ = isopropyl; **5a** R_1_ = H, R_2_ = OMe, R_3_ = H; **5b** R_1_ = H, R_2_ = isopropyl, R_3_ = H; **5c** R_1_ = H, R_2_ = H, R_3_ = isopropyl; **5d** R_1_ = H, R_2_ = OH, R_3_ = H; **5e** R_1_ = OH, R_2_ = H, R_3_ = Me; **5f** R_1_ = OH, R_2_ = H, R_3_ = *n*-pentyl. See Experimental part for synthetic details.

**Table 1 molecules-26-03944-t001:** Antileishmanial and antitrypanosomal activity pulchrol derivatives, compared to the positive controls benznidazole and miltefosine. See experimental for details about the assays.

Mol.	R_1_	R_2_	R_3_	R_6_′	R_6_′	*T. cruzi*	*L. braziliensis*	*L. amazonensis*
						IC_50_ (μM)	IC_50_ (μM)	IC_50_ (μM)
**1a**	H	OMe	Me	Me	Me	18.5 ± 9.6	59.2 ± 11.8	77.7 ± 5.6
**2b**	H	*i*-Pr	H	Me	Me	10.7 ± 4.3	12.1 ± 4.6	11.4 ± 3.6
**2c**	H	H	*i*-Pr	Me	Me	7.1 ± 1.4	17.8 ± 1.8	17.8 ± 0.7
**2d**	H	OMe	H	Me	H	125.8 ± 7.9	70.8 ± 19.7	44.0 ± 1.6
**2e**	H	OMe	H	H	Me	170.3 ± 7.9	118.0 ± 0.8	80.6 ± 4.7
**3b**	H	*i*-Pr	H	Me	Me	3.4 ± 0.2	8.8 ± 1.0	9.5 ± 4.1
**4b**	H	*i*-Pr	H	Me	Me	10.9 ± 3.8	272.9 ± 0.00	25.9 ± 5.5
**4c**	H	H	*i*-Pr	Me	Me	13.6 ± 5.7	63.3 ± 4.4	43.7 ± 8.2
**5b**	H	*i*-Pr	H	Me	Me	23.7 ± 8.6	49.2 ± 15.0	49.6 ± 4.5
**5c**	H	H	*i*-Pr	Me	Me	50.3 ± 11.3	311.6 ± 50.3	236.5 ± 48.8
**5d**	H	OH	H	Me	Me	54.9 ± 0.2	30.4 ± 2.9	33.3 ± 5.4
**5e**	OH	H	Me	Me	Me	5.9 ± 2.0	15.7 ± 5.1	21.2 ± 2.4
**5f**	OH	H	*n*-Pen	Me	Me	7.4 ± 0.6	10.3 ± 0.6	14.2 ± 1.3
	Benznidazole					19.2 ± 7.7		
	Miltefosine						13.0 ± 1.2	10.8 ± 1.5

**Table 2 molecules-26-03944-t002:** Proton chemical shifts (in ppm) for the compounds prepared in this study, measured at 400 MHz. The assignments were made with 2D NMR spectroscopy, COSY, HMQC and HMBC experiments.

Compd.	1-H	2-H	3-H	4-H	7-H	8-H	10-H	1′-H/H_2_/H_3_	2-OCH_3_	6-H/H_2_	6,6-CH_3_
**1a**	7.26	-	6.81	6.89	7.23	7.30	7.68	4.74	3.85	-	1.61
**2b** ^a^	7.65	-	7.15	6.90	7.41	7.79	8.24	10.07	-	-	1.65
**2c** ^b^	7.73	6.94	-	6.85	7.40	7.77	8.19	10.05	-	-	1.66
**2d**	7.31	-	6.87	6.94	7.34	7.81	8.17	10.07	3.87	5.25	1.63
**2e**	7.32	-	6.87	6.94	7.34	7.81	8.17	10.07	3.87	5.27	1.63
**3b** ^c^	7.65	-	7.13	6.89	7.32	7.85	8.32	-	-	-	1.64
**4b** ^d^	7.57	-	7.11	6.88	7.23	7.28	7.71	5.17	-	-	1.62
**4c** ^e^	7.64	6.90	-	6.82	7.22	7.24	7.66	5.14	-	-	1.63
**5b** ^f^	7.59	-	7.10	6.89	7.15	7.12	7.58	2.43	-	-	1.63
**5c** ^g^	7.63	6.88	-	6.82	7.12	7.08	7.51	2.39	-	-	1.62
**5d**	7.20	-	6.70	6.83	7.12	7.12	7.45	2.39	-	-	1.60
**5e** ^h^	-	6.28	-	6.43	7.15	7.07	8.17	2.39	-	-	1.60
**5f** ^i^	-	6.29	-	6.43	7.14	7.07	8.16	2.38	-	-	1.59

^a^ Isopropyl signals at 2.95 and 1.30 ppm. ^b^ Isopropyl signals at 2.90 and 1.27 ppm. ^c^ Methyl signal at 2.66 ppm and isopropyl signals at 2.95 and 1.30 ppm. ^d^ i-Butyl signals at 2.28, 2.16 and 0.99 ppm; and isopropyl signals at 2.94 and 1.30 ppm. ^e^ i-Butyl signals at 2.26, 2.15 and 0.97 ppm; and isopropyl signals at 2.88 and 1.26 ppm. ^f^ Isopropyl signals at 2.95 and 1.31 ppm. ^g^ Isopropyl signals at 2.88 and 1.26 ppm. ^h^ Methyl signal at 2.26 ppm. ^i^ *n*-Pentyl signals at 2.50, 1.61, 1.33, 1.31, 0.88 ppm.

**Table 3 molecules-26-03944-t003:** ^13^C NMR chemical shifts (in ppm) for compounds of series **1** to **5** determined at 100 MHz in CDCl_3_. The assignments were made with 2D NMR spectroscopy, COSY, HMQC and HMBC experiments.

Compd.	C-1	C-2	C-3	C-4	C-4a	C-6	C-6a	C-7	C-8	C-9	C-10	C-10a	C-10b	C-1′	2-OCH_3_	6,6-CH_3_/6-CH_3_
**1a**	108.0	154.6	115.5	118.8	146.9	77.4	139.5	123.7	126.8	140.4	121.0	129.1	123.0	65.3	56.0	27.5
**2b** ^a^	121.1	142.5	128.6	118.1	150.9	77.4	146.0	124.2	129.5	135.9	123.2	130.3	121.0	192.2	-	27.5
**2c** ^b^	123.1	120.5	152.2	115.9	152.8	77.7	145.4	124.2	129.0	136.0	123.2	130.2	118.9	192.1	-	27.6
**2d**	108.0	155.0	116.7	118.8	147.5	73.6	142.5	125.1	129.8	136.4	123.1	130.8	122.1	192.1	56.0	19.9
**2e**	108.0	155.1	116.7	118.9	147.5	73.6	142.6	125.1	129.8	136.4	123.1	130.8	122.1	192.1	56.0	19.9
**3b** ^c^	121.1	142.4	128.2	118.0	150.9	77.4	144.6	123.7	128.0	136.6	122.0	129.7	121.3	198.0	-	27.5
**4b** ^d^	120.8	142.0	127.8	117.9	150.9	77.4	139.7	123.7	127.7	135.6	122.1	129.4	121.7	66.0	-	27.7
**4c** ^e^	122.8	120.1	151.4	115.8	152.9	77.6	139.2	123.6	127.4	135.6	121.9	129.3	119.7	66.0	-	27.8
**5b** ^f^	120.7	141.9	127.4	117.8	151.0	77.4	137.1	123.3	128.6	137.2	122.8	127.7	122.1	21.5	-	27.8
**5c** ^g^	122.7	119.9	150.9	115.8	152.9	77.6	136.5	123.2	128.3	137.2	122.6	128.7	120.1	64.8	-	27.9
**5d**	109.5	150.1	116.3	118.9	146.9	77.4	137.4	123.3	129.1	137.3	123.1	128.4	123.5	21.4	-	27.6
**5e** ^h^	153.2	110.7	139.5	111.6	154.8	77.4	137.0	122.8	127.7	137.0	126.6	127.6	108.7	21.7	-	27.2
**5f** ^i^	153.2	110.0	144.7	110.9	154.8	77.4	137.0	122.8	127.7	127.7	126.5	137.0	108.8	21.7	-	27.2

^a^ Isopropyl signals at 33.9 and 24.4 ppm. ^b^ Isopropyl signals at 34.2 and 23.9 ppm. ^c^ Methyl signal at 26.9 ppm and isopropyl signals at 33.9 and 24.4 ppm. ^d^ 3-Methylbutanoate signals at 173.1, 43.6, 25.9 and 22.6 ppm and isopropyl signals at 33.8 and 24.4 ppm. ^e^ 3-Methylbutanoate signals at 173.1, 43.6, 25.9 and 22.6 ppm and isopropyl signals at 34.1 and 23.9 ppm. ^f^ Isopropyl signals at 33.9 and 24.4 ppm. ^g^ Isopropyl signals at 34.1 and 24.0 ppm. ^h^ Methyl signal at 21.4 ppm. ^i^ *n*-Pentyl signals at 35.8, 31.6, 30.6, 22.7 and 14.2 ppm.

## Data Availability

All data are available in this publication and in the Appendix A.

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
