# Peer review of "SARs for the Antiparasitic Plant Metabolite Pulchrol. 3. Combinations of New Substituents in A/B-Rings and A/C-Rings"

_molecules, 2021, doi:10.3390/molecules26133944_

Round 1
Reviewer 1 Report
The manuscript "SARs for the antiparasitic plant metabolite pulchrol. Combinations of new substituents in A/B-rings and A/C-rings"
is an interesting study of SAR based on natural compounds scaffold of active molecule pulchrol.
The first comment is about the information that calculates SI based on macrophage Raw cells. Generally, the cytotoxicity in these cell types is only determined to establish the concentration that will use to test intracellular amastigotes of Leishmania, for example. The specialists in trypanosomatids, in agreement with notes of the World Health Organization, recommend the cytotoxicity in other cell lineages that affect more intensively the host of these parasites.
I think that the result of cytotoxicity must be shown in this manuscript, but to improved the model of study of cytotoxicity and quality of this manuscript the cytotoxicity must be determined in the L6 cells or other models.
I understand that the new experiments at this point can be hard for the authors and I recommend strongly that the SI can be removed. If the authors agree with that, the discussion of the cytotoxicity must be direct to the future study of intracellular amastigotes using macrophage RAW lineage as a model of Leishmania infection.
Figure 1. I think that the Benzo[c]chromone scaffold that the legend referred to as represent in the "left" is the same structure of compound 1a (pulchrol structure). Please, revise the legend to be clear.
I would like to view a figure that resumes the main SARs conclusion and also suggest that the comparison of activity between the compounds can be shown in a scheme showing the structures to highlighting the differences.
The section conclusion contains elements of the discussion section and must be reviewed. Please, the conclusion must be shorter: I suggest one paragraph with 10-15 lines.
Finally, I suggest the separation of sections "Results" and "Discussion" to improve the manuscript.
Author Response
Sirs,
I hereby resubmit the manuscript molecules-1174607:
"SARs for the antiparasitic plant metabolite pulchrol. Combinations of new substituents in A/B-rings and A/C-rings"
by Paola Terrazas , Efrain Salamanca , Marcelo Dávila , Sophie Manner, Alberto Gimenez , and Olov Sterner,
for reconsideration for publication in Molecules. I am sorry that it has taken so long for me to do this revision, but I have been severely ill. I feel better now.
Lund June 12th, 2021
Olov Sterner
Responses to the reviewers comments:
Reviewer 1:
- I fully agree with the reviewer, the choice of a suitable cell line for establishing the cytotoxicity and thereby establish the potential usefulness of a antiparasitic effect is very important. However, as an academic laboratory we do not have a free choice, but are restricted to what is available. In addition, the macrophage Raw cells were used in the previous and published studies (references 44 and 45), and any comparisons would be facilitated by the use of the same cell line. For this reason we would also like to keep the SI-values. Our next step is to use the QSARs suggested in this manuscript to design and prepare new derivatives that would be expected to possess potent antiparasitic effects, or be inactive, and the remarks by the reviewer will prompt us to expand the cytotoxicity assays to include other cell lines.
- Figure 1, I have changed the Figure as well as the legend as suggested, in order to be more clear.
- The SARs suggested are discussed extensively in the text, and a figure that summarizes them is planned for the next manuscript (the design and preparation of new derivatives that would be expected to be active or inactive). At this stage any SARs are only suggestions that need to be confirmed, and I believe that one should be careful with how they are presented.
- The section "Conclusions" has been modified as suggested.
- The sections "Results " and "Discussion" have been separated as suggested.
Reviewer 2:
- I have gone through the manuscript in detail, and corrected it.
- The clarifications and corrections have been made as suggested.
- The clarifications and corrections have been made as suggested.
Reviewer 2 Report
This manuscript by Terrazas et al. describes the derivation of various analogues of a natural compound, pulchrol, and analysis of their efficacy against various trypanosomatid species. The manuscript is generally well written and easy to follow. The findings are significant and interesting. However, a few issues need to be addressed as follows:
- There are numerous grammatical errors that need to be fixed by careful reading of the manuscript in order to make it easier for the reader to understand.
- Section 3.3.1: Contains unqualified abbreviations (what is “PMS” and “XTT”?). Describe how the IC50 values were derived. The parasite number should read 3 x 10e6.
- Section 3.3.2. Describe the recipe of LIT medium. The parasite number should read 3 x 10e6.
Author Response

(The authors gave the same response as above.)

Round 2
Reviewer 1 Report
The revision of the manuscript was improved and the majority of suggestions addressed by the authors.
The exception is the maintenance of SI calculation using promastigotes and Raw macrophage lineage. The cytotoxicity in macrophages is only estimated to test the intracellular amastigotes activity of drug candidates. The SI showed here is lower than 1 for the most of compounds and did not show the reality of toxicity against somatic cells.
The SI comparing macrophages and intracellular amastigotes could be accepted, but this data is not available in this study.
Author Response
The results from the cytotoxicity on Raw macrophages has been deleted, and consequently also the SI-values. Only the antiparasitic activity is reported and discussed.